# A Large Scale Search Dataset for Unbiased Learning to Rank

**Lixin Zou**[1*], **Haitao Mao**[2*†], **Xiaokai Chu**[1†], **Jiliang Tang**[2],
**Shuaiqiang Wang**[1], **Wenwen Ye**[1], **Dawei Yin**[1‡]
[1]Baidu Inc., [2]Michigan State University
{zoulixin15,xiaokaichu, two_ye, shqiang.wang}@gmail.com
{haitaoma,tangjili}@msu.edu, yindawei@acm.org

## Abstract

The unbiased learning to rank (ULTR) problem has been greatly advanced by recent deep learning techniques and well-designed debias algorithms. However, promising results on the existing benchmark datasets may not be extended to the practical scenario due to some limitations of existing datasets. First, their semantic feature extractions are outdated while state-of-the-art large-scale pre-trained language models like BERT cannot be utilized due to the lack of original text. Second, display features are incomplete; thus in-depth study on ULTR is impossible such as the displayed abstract for analyzing the click necessary bias. Third, synthetic user feedback has been adopted by most existing datasets and real-world user feedback is greatly missing. To overcome these disadvantages, we introduce the Baidu-ULTR dataset. It involves randomly sampled 1.2 billion searching sessions and 7,008 expert annotated queries (397,572 query document pairs). Baidu-ULTR is the first billion-level dataset for ULTR. Particularly, it offers: (1) the original semantic features and pre-trained language models of different sizes; (2) sufficient display information such as position, displayed height, and displayed abstract, enabling the comprehensive study of multiple displayed biases; and (3) rich user feedback on search result pages (SERPs) like dwelling time, allowing for user engagement optimization and promoting the exploration of multi-task learning in ULTR. Furthermore, we present the design principle of Baidu-ULTR and the performance of representative ULTR algorithms on Baidu-ULTR. The Baidu-ULTR dataset and corresponding baseline implementations are available at `https://github.com/ChuXiaokai/baidu_ultr_dataset`. The dataset homepage is available at `https://searchscience.baidu.com/dataset.html`.

## 1 Introduction

Learning to Rank (LTR) that aims to measure documents' relevance w.r.t. queries is a popular research topic with applications in web search engines, e-commerce, and multiple different streaming services [22]. With the rise of deep learning, the heavy burden of data annotation drives the academia and industry communities to the study of learning to rank using implicit user feedback (e.g., user click). However, directly optimizing the model with click data results in unsatisfied performance due to the existence of biases, such as position bias [18], trust bias [1], and click necessary bias [21]. Unbiased learning to rank (ULTR) has been proposed for mitigating biases in user feedback with counterfactual learning algorithms [18, 29, 30]. To meet the demand of ULTR, numerous datasets have

---

*Equal contribution.

†Work conducted during an internship at Baidu Inc.

‡Corresponding author.

36th Conference on Neural Information Processing Systems (NeurIPS 2022).

been released publicly such as Yahoo! LETOR [7], Microsoft LETOR [22], Istella LETOR [4] [10], and Tiangong-ULTR [5] [2, 3]. These datasets include thousands of annotated queries, hundreds of semantic features, and millions of user sessions (from Tiangong), which have been extensively adopted for the ULTR research [4].

Although existing ULTR datasets can meet the data-consuming requirement to some extent, there are still some limitations. First, the provided semantic features (e.g., BM25 [24, 25, 23], TF-IDF [26], LMABS [33]) cannot enjoy the advantages of modern representation learning techniques. Applying the recent paradigm of large-scale pre-training and end-to-end finetuning (e.g., RoBERTa [19], ERNIE [34], Poly-Encoder[15]) requires the access to the original text of queries and documents. Second, limited types of display information are provided. The ranking position from the Tiangong-ULTR dataset is the only publicly available display information. As expected, position-related biases (e.g., position bias [18], trust bias [1]) have been widely studied. However, biases related to other display information are overlooked. For example, users unnecessarily click the document if the document displayed abstract can perfectly meet user requirement [21]. Moreover, documents with different multimedia types could have a distinct attraction to users, e.g., videos and pictures are more attractive than plain text. Third, most existing datasets lack real-world user feedback. Though Tiangong-ULTR provides real-world click data, a small test set with only 100 queries makes it difficult to produce reliable and significant results. Alternatively, the research community seeks to simulate click data with the clicking behavior assumption that is consistent with the proposed methods [2, 17, 16]. Though significant improvement has been observed in simulations, such success is hard to be extended to the practical scenario [4, 2, 8].

To address the aforementioned challenges, we introduce the Baidu-ULTR dataset, a large-scale unbiased learning to rank dataset for web search. It randomly sampled 1.2 billion searching sessions from the largest Chinese search engine – Baidu, and 7,008 expert annotated queries (397,572 query document pairs) for validation and test. Overall, Baidu-ULTR has the following advantages over existing ULTR datasets:

- Baidu-ULTR provides the original text of queries and documents after desensibilisation. It enables us to construct both handcraft features and semantic features generated by advanced language models. In addition, we provide advanced language models pretrained with the MLM loss [11].
- Baidu-ULTR offers diverse display information such as position, displayed height, and so on. It enables the study of multiple biases with advanced techniques like causal discovery.
- Rich user behaviors, e.g., click, skip, dwelling time, and displayed time, have been recorded, offering opportunities for optimizing user engagement and exploring multi-task learning in ULTR.
- Baidu-ULTR is a large-scale web search dataset (1.2 billion searching sessions) with sufficient expert annotations (397,572 expert-annotated query document pairs), which further supports pre-training large-scale language models, and studies on the pre-training task for web search.
- Baidu-ULTR introduces a more practical scenario with significant challenges in ULTR like ranking with long-tail queries and conquering the mismatch between training and test sets.

The rest of the paper is organized as follows. In Section 2, we briefly review the ULTR task and existing datasets. In Section 3, we give a detailed introduction on the dataset collection and data analysis. In Section 4, we conduct empirical studies with state-of-the-art ULTR methods. Finally, we discuss the impact and potential limitations of Baidu-ULTR in Section 5.

## 2 Preliminary

In this section, we briefly introduce the ULTR task and review the existing ULTR datasets with a detailed comparison with Baidu-ULTR in Tab. 3.

### 2.1 Unbiased Learning to Rank

The task of ranking is to measure the relative order among a set of $N$ documents $\mathcal{D}_q = \{d_i\}_{i=1}^{N}$ under the constraint of a query $q \in \mathcal{Q}$, where $\mathcal{D}_q \subset \mathcal{D}$ is the set of $q$-related documents retrieved from all

---

[4]http://quickrank.isti.cnr.it/istella-dataset/
[5]http://www.thuir.cn/data-tiangong-ultr/

indexed documents $\mathcal{D}$, and $\mathcal{Q}$ is the set of all possible queries. We aim to design a scoring function $f(q, d) : \mathcal{Q} \times \mathcal{D} \to \mathbb{R}$ that descendingly sorts the documents as a list $\pi_{f,q}$ to maximize an evaluation metric $\vartheta$ (e.g., DCG [16], PNR [35], and ERR [7]) as

$$f^* = \max_f \mathbb{E}_{q \in \mathcal{Q}} \vartheta(\mathcal{R}_q, \pi_{f,q}). \tag{1}$$

where $\mathcal{R}_q = \{r_d\}_{d \in \mathcal{D}_q}$ is a set of relevance labels $r_d$ corresponding to $q, d$. Usually, $r_d$ is the graded relevance in 0-4 ratings, which indicates the relevance of document $d_i$ as {**bad**, **fair**, **good**, **excellent**, **perfect**}, respectively. To learn the scoring function $f$, a corresponding loss function is needed to approximate $r_d$ with $f(q, d)$ as

$$\ell_{ideal}(f) = \mathbb{E}_{q \in \mathcal{Q}} \left[ \sum_{d \in \mathcal{D}_q} \Delta(f(q, d), r_d) \right], \tag{2}$$

where $\Delta$ is a function that computes the individual loss for each document. If all the documents are annotated, $\ell_{ideal}(f)$ would be the ideal ranking loss for optimizing the ranking function. Relevance annotation $r_d$ is elicited by expert judgment; thus $r_d$ is considered to be unbiased, but expensive.

An alternative but intuitive approach is to use the user's implicit feedback as the relevance label. For example, by replacing the relevance label $r_d$ with click label $c_d$ in Equ. 2, a naive empirical ranking loss is derived as follows:

$$\ell_{naive}(f) = \frac{1}{|\mathcal{Q}_o|} \sum_{q \in \mathcal{Q}_o} \left[ \sum_{d \in \mathcal{D}_q} \Delta(f(q, d), c_d) \right], \tag{3}$$

where $\mathcal{Q}_o$ is the *observed* query set. $c_d$ is a binary variable indicating whether the document $d$ in the ranked list is clicked or not. However, this naive loss function is biased since the display of the final ranking result may influence user's click [31]. For instance, position bias occurs because users are more likely to examine the documents at higher ranks [18]. Consequently, highly ranked documents may receive more clicks, and relevant (but unclicked) documents may be perceived as negative samples because they are unexamined by users. To address this issue, unbiased learning to rank has been proposed to remove the effect of data bias in the computation of the ranking loss $\ell_{naive}(f)$. so that the model trained with biased data (e.g., clicks) would converge to that trained with unbiased labels (i.e., the relevance of a document).

Table 1: Characteristics of publicly available datasets for unbiased learning to rank.

| | | Training Implicit Feedback Data | | | | Validation & Test Data | | | | |
| --- | --- | --- | --- | --- | --- | --- | --- | --- | --- | --- |
| Dataset | # Query | # Doc | # User Feedback | # Display-info | # Session | # Query | # Doc | # Label | # Feature | Pub-Year |
| Yahoo Set1 | 19,944 | 473,134 | 1 (Simulated click) | 1 (Position) | - | 9,976 | 236,743 | 5 | 519 | 2010 |
| Yahoo Set2 | 1,266 | 34,815 | 1 (Simulated click) | 1 (Position) | - | 5,064 | 138,005 | 5 | 596 | 2010 |
| Microsoft | $\approx$18,900 | $\approx$2,261,000 | 1 (Simulated click) | 1 (Position) | - | $\approx$12,600 | $\approx$1,509,000 | 5 | 136 | 2010 |
| Istella | 23,219 | 7,325,625 | 1 (Simulated click) | 1 (Position) | - | 1,559 | 550,337 | 5 | 220 | 2016 |
| Tiangong | 3,449 | 333,813 | 1 (Real Click) | 1 (Position) | 3,268,177 | 100 | 10,000 | 5 | 33 | 2018 |
| Baidu | 383,429,526 | 1,287,710,306 | 18 (Real Feedback) | 8 (Display Info) | 1,210,257,130 | 7,008 | 367,262 | 5 | ori-text | 2022 |

## 2.2 Existing ULTR Datasets

Existing publicly available ULTR datasets can be roughly categorized by utilizing synthetic or real user feedback. Both have been widely adopted by the empirical study of ULTR algorithms.

**Synthetic Data** Yahoo! LETOR [7], Microsoft LETOR [22] and Istella LETOR [10] are three commonly used datasets with synthetic user feedback. Notice that Yahoo! LETOR includes two sets, Yahoo Set1, and Yahoo Set2. Due to privacy concerns, those datasets do not release real user feedback and hide the original text of queries and documents. Thus, researchers have to simulate click data following a specific user behavior assumption, such as position-dependent click model [18], and cascade click model [9]. Furthermore, a set of predefined semantic features is used to represent the original queries and documents. Detailed statistics about those datasets are illustrated in Tab. 3.

**Real Data** The Tiangong-ULTR [2, 3] is the only dataset with real-world user feedback supporting the research of unbiased learning to rank. It provides real-world click data sampled from the search

sessions of Sogou.com [6] for training and a separate expert annotated test set for the performance evaluation. However, the test set with only 100 queries is insufficient to draw a significant conclusion with the dataset. Detailed statistics about Tiangong-ULTR can be found in Tab. 3.

Table 2: Characteristics of publicly available datasets for unbiased learning to rank.

| Dataset | # Query | # Doc | # User Feedback | # Display-info | # Session |
|---|---|---|---|---|---|
| Yahoo Set1 | 19,944 | 473,134 | 1 (Simulated click) | 1 (Position) | - |
| Yahoo Set2 | 1,266 | 34,815 | 1 (Simulated click) | 1 (Position) | - |
| Microsoft | ≈18,900 | ≈2,261,000 | 1 (Simulated click) | 1 (Position) | - |
| Istella | 23,219 | 7,325,625 | 1 (Simulated click) | 1 (Position) | - |
| Tiangong | 3,449 | 333,813 | 1 (Real Click) | 1 (Position) | 3,268,177 |
| Baidu | 383,429,526 | 1,287,710,306 | 18 (Real Feedback) | 8 (Display Info) | 1,210,257,130 |

Table 3: Characteristics of publicly available datasets for unbiased learning to rank.

| Dataset | # Query | # Doc | # Label | # Feature | Pub-Year |
|---|---|---|---|---|---|
| Yahoo Set1 | 9,976 | 236,743 | 5 | 519 | 2010 |
| Yahoo Set2 | 5,064 | 138,005 | 5 | 596 | 2010 |
| Microsoft | ≈12,600 | ≈1,509,000 | 5 | 136 | 2010 |
| Istella | 1,559 | 550,337 | 5 | 220 | 2016 |
| Tiangong | 100 | 10,000 | 5 | 33 | 2018 |
| Baidu | 7,008 | 367,262 | 5 | ori-text | 2022 |

## 3 Dataset Description

In this section, we formally introduce the Baidu-ULTR dataset. It consists of two parts: **(1)** Large Scale Web Search Sessions and **(2)** Expert Annotation Dataset. Next, we will first detail these two parts. Then, we provide detailed data analysis in Section 3.3 for a better understanding of the collected dataset. The data license is further provided in Section 3.4.

### 3.1 Large Scale Web Search Sessions

**Queries** Queries are randomly sampled from search sessions of the Baidu search engine in April 2022. Each query is given a unique identifier. A frequent query is likely to be sampled several times from search sessions, and each replicate is given a different identifier. This ensures the query distribution in Baidu-ULTR follows the same distribution as that in the online system: frequent queries have larger weights.

**Documents** For the candidate documents of the query, we only record the displayed documents to save the cost of storing the logs. Typically, the document not being displayed is less informative than the ones displayed since the user cannot provide any feedback on the document without display. As a result, the logged search session usually contains just 10 documents for every query because one page contains 10 results. Only 1.1% search sessions contain over 10 displayed documents, i.e., only 1.1% of users turn the page. This phenomenon further results in the mismatch between the training and test. Specifically, in the training phase, users just "label" the top-10 results. However, in the inference stage, the top-10 results are generated through **retrieval** and **ranking**, where the systems are required to rank billion documents or thousands of documents by multiple models such as Inverted index [32], BM25 [23], TF-IDF [5], Bi-Encoder [15], RankNet [6], BM25, DSSM [28] and Cross-encoder [35]).

**Page Presentation Features** User behaviors on search result pages (SERPs) are highly biased by the layout of SERPs [31], such as the position (position bias [18]), the displayed abstract of the document (click necessary bias [21]), and the displayed area of the document. To support advanced research in ULTR, we collect the following rich presentation information of the document on SERPs: the ranking **position**, the displayed **url**, the displayed **title** of the document, the displayed **abstract** of the document, the **multimedia type** of the document, and the height of SERP (i.e., the vertical pixels of SERP on the screen). An illustration of the page presentation features is shown in Fig. 1(a). The type and description of presentation features are illustrated in Tab. 8 in Appendix A.1.

---

[6]https://www.sogou.com/

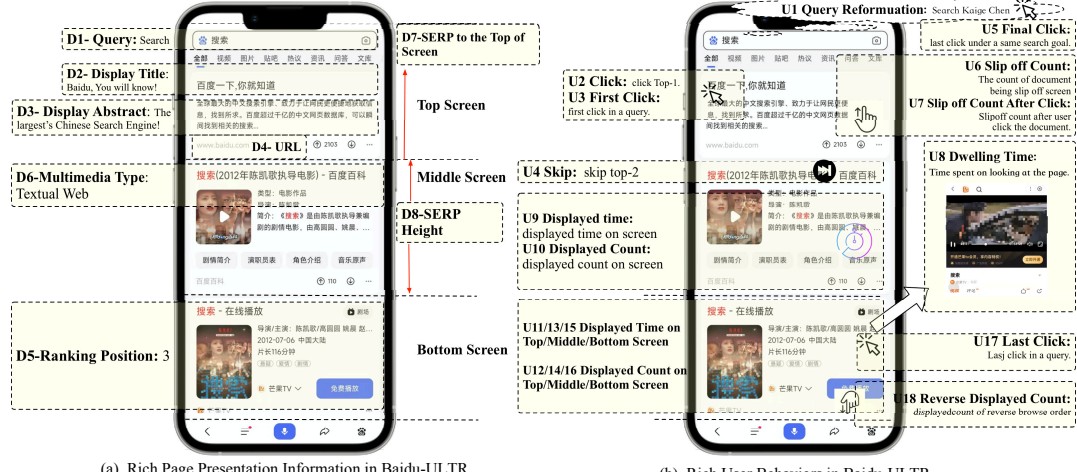

(a) Rich Page Presentation Information in Baidu-ULTR

(b) Rich User Behaviors in Baidu-ULTR

Figure 1: (a) A demo explanation of rich page presentation information in Baidu-ULTR. There are 8 presentation features that start from D1 to D8. Detailed descriptions are illustrated in Tab. 8 in Appendix A.1. (b) A demo explanation of rich user behaviors in Baidu-ULTR. There are 18 user behaviors starting from U1 to U18. Detailed descriptions are illustrated in Tab. 9 in Appendix A.1.

For user privacy protection, **the original texts of query, title and abstract are denoted as sequential token ids with a private dictionary**. To further prevent losing semantic information, we provide a unigram set that records the high-frequency words using the desensitization token ids, which is useful for modeling the word-level semantic information. For easy usage, we provide a set of pre-trained language models with various sizes (pre-trained with MLM and naive loss mentioned in Equ. 3) that can translate the raw text into dense semantic features.

**User Behaviors** Due to the lack of user behaviors, ULTR research mainly focuses on click modeling [4] and few industrial works analyze the user's satisfaction with rich user behaviors [31]. To facilitate the research on ULTR, we provide a set of rich users' behaviors, including user's **query reformulation**, the **skip**, the **click**, the **first click**, the **last click**, user's **dwelling time**, the **displayed time** on the screen, the **displayed count** on the screen, the **slip off count**, and the displayed count of reverse browsing. A demo explanation is presented in Fig. 1(b). The type and description of user behaviors are illustrated in Tab. 9 in Appendix A.1.

## 3.2 Expert Annotation Dataset

**Queries** Queries are randomly sampled from the monthly collected query sessions of the Baidu search engine, which is similar to the query collection in Section 3.1. Since the search queries are heavy-tailed distributed (depicted in Fig. 2(a)), we further provide the frequency of queries. Specifically, queries are descendingly split into 10 buckets according to their monthly search frequencies, where buckets 0, 1, 2, buckets 3, 4, 5, 6, and buckets 7, 8, 9 correspond to the high-frequency, mid-frequency, tail frequency, respectively. The rankers' performance w.r.t. the queries frequency can be further analyzed, which is beneficial for understanding the model's strengths and weaknesses under the long-tail phenomenon.

**Documents** To simulate the online scenario, candidate documents are selected from the retrieval phase. Specifically, we record the top-30 documents from the retrieval model as the candidate set for ranking. We further cover the top thousands of results from the retrieval stage by selecting the document with an interval of 30 (i.e., the documents ranked at position

Table 4: Distribution of Relevance Labels.

| Grade | Label | # Query-Doc | Ratio of Label |
|---|---|---|---|
| Perfect | 4 | 714 | 1.80% |
| Excellent | 3 | 28,172 | 9.21% |
| Good | 2 | 112,759 | 28.36% |
| Fair | 1 | 36,622 | 9.21% |
| Bad | 0 | 219,305 | 55.16% |

$\{30, 60, 90, 120, 150, ..., 990\}$). These documents are used to measure rankers' performance on distinguishing documents that are extremely disruptive to the user experience.

**Expert Annotation** The relevance of each document to the query has been judged by expert annotators who assign one of 5 labels, {**bad**, **fair**, **good**, **excellent**, **perfect**} to the document. Each of these

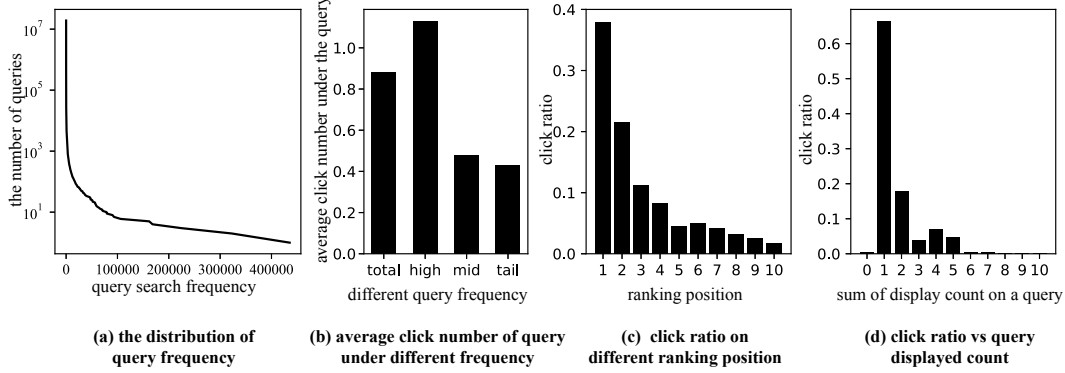

(a) the distribution of query frequency    (b) average click number of query under different frequency    (c) click ratio on different ranking position    (d) click ratio vs query displayed count

Figure 2: Data analysis on Baidu-ULTR: (a) the distribution of query frequency. (b) the average click number of queries under different frequencies. (c) the click ratio on different ranking positions. (d) the click ratio vs the query displayed count (the sum of display count on a query).

relevance labels is then converted to an integer ranging from 0 (for bad) to 4 (for perfect). Specific guidelines, depicted in Tab. 5, are given to expert annotators for instructing them on how to perform relevant judgments. The main purpose of these guidelines is to reduce the amount of disagreement across expert annotators. Additionally, only the label that most expert annotators agree on will be selected as the final label. Tab. 4 illustrates the distribution of the relevance labels. We can observe: **(1)** perfect only occupies 1.8% in all expert annotations since a perfect will be only given to the destination page of a navigational query according to Tab. 5; **(2)** the bad documents take over 50% of documents. The reason is that the irrelevant documents are usually the majority for long-tail queries, e.g., "Does the China Tobacco Bureau have any temporary job position?" Unfortunately, the long-tail queries are also the majority for the search engine (referred to Fig. 2(a)).

Table 5: The general guideline of annotation.

| Label | Guideline |
|---|---|
| 0 (bad) | Useless or outdated documents that do not meet the requirements at all. |
| 1 (fair) | Helpful to some extent but deficient in authority, timeliness document. |
| 2 (good) | Meet the requirement of the query. |
| 3 (excellent) | Meet the requirement of the query and timeliness document. |
| 4 (Perfect) | Meet the requirement of the query, timeliness, and authoritative document. |

### 3.3 Dataset Analysis

In this subsection, we present our primary data analysis on the Baidu-ULTR dataset. A full analysis procedure and more detailed results can be found in Appendix B. As shown in Fig. 2, we have the following observations:

1. Long-tail distribution appears in many user behaviors. As shown in Fig. 2(a), over 60% searches are based on top 10% high-frequency queries, while most queries only appear very few times. This phenomenon also happens on the displayed count, the displayed time, the number of clicks per query, and the number of skips per query (depicted in Appendix B). The long tail distribution can significantly affect the ULTR performance. As reported in Tab. 7, all ULTR algorithms perform poorly on those tail queries.

2. The logged search results on the high-frequency queries are more relevant than those on tail queries since the number of clicks per query on the high-frequency queries with an average click of 1.13 is much larger than those on tail queries with an average click of 0.43 (shown in Fig. 2(b)).

3. The most critical user feedback (i.e., click) shows a strong correlation with both page presentation features, e.g., position, and other user behaviors, e.g, displayed count. We

illustrate the click ratio on different ranking positions in Fig. 2(c). The click ratio decreases as the documents are displayed in the latter positions. Similar observations can also be made in click and displayed count. As shown in Fig. 2(d), more than 70% of clicks are done with only one displayed count on the document belonging to the query.

4. As the displayed time becomes longer, users are more likely to spend more time on the top of the screen, from 25% to nearly 35%, while less time is spent on the bottom of the screen from more than 40% to 30% (depicted in Fig. 6).

## 3.4  Baidu-ULTR License

The dataset can be freely downloaded at `https://github.com/ChuXiaokai/baidu_ultr_dataset` and noncommercially used with a custom license CC BY-NC 4.0[7]. Besides the current tasks in the dataset directory, users can define their own ones under the license.

## 4  Benchmark and Baselines

In this section, we conduct an empirical study of several benchmark unbiased learning to rank algorithms on Baidu-ULTR and further present their performance versus different query frequencies.

### 4.1  Baseline Methods

To fully test unbiased learning to rank algorithms with different learning paradigms, we select the following representative baselines:

- **Naive**: It directly trains the model with user feedback without any correction, as stated in Equ. 3.
- **IPW**: Inverse Propensity Weighting is one of the first ULTR algorithms proposed under the framework of counterfactual learning [18], which weights the training loss with the probability of the document being examined in the search session.
- **DLA**: The Dual Learning Algorithm [2] treats the problem of unbiased learning to rank and unbiased propensity estimation as a dual problem, such that they can be optimized simultaneously.
- **REM**: The Regression EM model [30] uses an EM framework to estimate the propensity scores and ranking scores.
- **PairD**: The Pairwise Debiasing Model [14] uses inverse propensity weighting for pairwise learning to rank.

Notably, all the above ULTR algorithms only take the position-related bias into consideration without utilizing other display features. Moreover, they only include the click data as the supervised signal without any other user behavior data. Therefore, there are great potentials for the design of the new ULTR algorithms on Baidu-ULTR dataset by utilizing the new extracted SERPs and display features.

### 4.2  Metrics

The following evaluation metrics are employed to assess the performance of the ranking system. The **Discounted Cumulative Gain** (DCG) [16] is a standard listwise accuracy metric and is widely adopted in the context of ad-hoc retrieval. For a ranked list of $N$ documents, we use the following implementation of DCG:

$$DCG@N = \sum_{i=1}^{N} \frac{G_i}{\log_2(i+1)},$$

where $G_i$ represents the weight assigned to the document's label at position $i$. A higher degree of relevance corresponds to a higher weight. We use the symbol $DCG$ to indicate the average value of

---

[7] `https://creativecommons.org/licenses/by-nc/4.0/`

this metric over the test queries. $DCG$ will be reported only when absolute relevance judgments are available.

The **Expected Reciprocal Rank** (ERR) [13] calculates the expectation of the reciprocal of the position of a result at which a user stops. This measure is defined as:

$$ERR@N = \sum_{i=1}^{N} \frac{1}{i} \prod_{j=1}^{i-1} \left(1 - R_j\right) R_i,$$

where $R_i$ indicates the relevance probability of the $i$-th document to the query and the expression $\frac{1}{i} \prod_{j=1}^{i-1} \left(1 - R_j\right)$ represents the non-relevance probability of the ordered documents prior to the position of the $i$-th document in the list.

### 4.3 Model Setup

In our experiments, we employ a transformer-based cross-encoder with 12 layers, 12 heads, and 768 hidden size as the backbone model for approximating the scoring function $f(q, d)$. For easy usage, we provide a warm-up model for initialization, which is trained with the mixture of the naive loss (Equ. 3) and the masked language modeling (MLM) loss [11] by randomly masking 10% tokens. With the pre-trained model, the ranking function is set by using the `[CLS]` embedding with 3-layers MLP (hidden layer of 512-256-128) and optimized with the Adam optimizer (learning rate = $2 \times 10^{-6}$). All the experiments are obtained by an average of 5 repeat runs. The hyper-parameters for training the ULTR algorithms are selected from ULTR toolkit[8] [27]. The expert annotation dataset is split into validation and test sets according to a 20%-80% scheme. All the models are trained on the machine with 28 Intel(R) 5117 CPU, $32G$ Memory, 8 NVIDIA V100 GPUs, and 12T Disk.

### 4.4 Performance Comparison

Tab. 6 summarizes the DCG and ERR performance of selected learning algorithms on the Baidu-ULTR dataset. From the table, we have the following observations: (1) No ULTR algorithms provide satisfying results since they only take the position-related biases into consideration. (2) The DLA algorithm performs best across all algorithms, showing its robustness on the real dataset. (3) The naive algorithm shows comparable performance with IPW, even better than REM and PairD. It reveals that real-world user feedback can be more complex than synthetic feedback generated with specific user behavior assumptions like position-dependent click model [18]. Therefore, ULTR algorithms with good performance on synthetic datasets may not show consistently good performance in the real-world scenario. An intuitive explanation is that the user logs are collected from a well-performed search engine, the click may not have such bias as mentioned in the assumption. Directly mitigating the bias may lead to over-mitigation and unsatisfying results.

Table 6: Comparison of unbiased learning to rank (ULTR) algorithms with different learning paradigms on Baidu-ULTR using cross-encoder as ranking models. The best performance is highlighted in boldface.

| | DCG@1 | ERR@1 | DCG@3 | ERR@3 | DCG@5 | ERR@5 | DCG@10 | ERR@10 |
|---|---|---|---|---|---|---|---|---|
| Naive | 1.235±0.029 | 0.077±0.002 | 2.743±0.072 | 0.133±0.003 | 3.889±0.087 | 0.156±0.003 | 6.170±0.124 | 0.178±0.003 |
| IPW | 1.239±0.038 | 0.077±0.002 | 2.742±0.076 | 0.133±0.003 | 3.896±0.100 | 0.156±0.004 | 6.194±0.115 | 0.178±0.003 |
| REM | 1.230±0.042 | 0.077±0.003 | 2.740±0.079 | 0.132±0.003 | 3.891±0.099 | 0.156±0.004 | 6.177±0.126 | 0.178±0.004 |
| PairD | 1.243±0.037 | 0.078±0.002 | 2.760±0.078 | 0.133±0.003 | 3.910±0.092 | 0.156±0.003 | 6.214±0.114 | 0.179±0.003 |
| DLA | **1.293**±0.015 | **0.081**±0.001 | **2.839**±0.011 | **0.137**±0.001 | **3.976**±0.007 | **0.160**±0.001 | **6.236**±0.017 | **0.181**±0.001 |

### 4.5 Performance Comparison on Tail Query

To investigate more on the effect of search frequencies, we further compare the ULTR algorithm performance versus three different levels of search frequencies, including high, middle, and tail. Results in Tab. 7 demonstrate the following observations: (1) performance of all the algorithms decreases progressively from high frequency queries to tail queries, which indicates the difficulty

---

[8]`https://github.com/ULTR-Community/ULTRA_pytorch`

of learning to rank on the tail queries; (2) The naive algorithms show competitive performance on the tail queries, revealing that the benchmark methods are fragile in dealing with the data bias in tail queries. (3) The DLA algorithm illustrates the considerable improvement in the high-frequency queries, suggesting that the ULTR algorithms are beneficial for the high-frequency ranking problem.

Table 7: Performance comparison of evaluation ULTR algorithms versus different search frequencies. The best performance is highlighted in boldface.

| Model | DCG@3 | | | DCG@5 | | | DCG@10 | | |
|---|---|---|---|---|---|---|---|---|---|
| | High | Mid | Tail | High | Mid | Tail | High | Mid | Tail |
| Naive | 3.960±0.058 | 2.992±0.119 | 1.742±0.079 | 5.596±0.098 | 4.254±0.142 | **2.474**±0.092 | 8.812±0.140 | **6.777**±0.173 | 3.942±0.121 |
| IPW | 4.017±0.132 | 2.976±0.111 | 1.722±0.061 | 5.699±0.145 | 4.235±0.140 | 2.447±0.090 | 8.969±0.146 | 6.762±0.163 | 3.925±0.109 |
| REM | 3.994±0.114 | 2.982±0.124 | 1.723±0.067 | 5.665±0.128 | 4.237±0.158 | 2.454±0.074 | 8.904±0.147 | 6.755±0.183 | 3.927±0.104 |
| PairD | 4.018±0.102 | 2.993±0.110 | **1.750**±0.079 | 5.662±0.120 | 4.267±0.129 | 2.474±0.088 | 8.924±0.145 | 6.804±0.153 | **3.961**±0.119 |
| DLA | **4.226**±0.042 | **3.073**±0.022 | **1.750**±0.016 | **5.894**±0.030 | **4.300**±0.020 | 2.472±0.009 | **9.147**±0.044 | 6.767±0.027 | 3.920±0.009 |

## 5  Discussion

In this section, we will discuss (1) the challenges of designing well-performed algorithms on our Baidu-ULTR; (2) the new research topics introduced to the academic study; (3) the potential limitations existing in Baidu-ULTR.

### 5.1  Data Challenges

In this subsection, we introduce three new challenges on how to debias in Baidu-ULTR, which is more practical with a close connection to the industry scenarios.

**Biases in Real-World User Feedback**    Different from other datasets, the user feedback and display information are collected from the user logs in Baidu search engine. As the user logs are collected from a well-performed search engine, the click may not have the same simple and strong bias as the assumption in generating synthetic datasets, i.e. [18, 2, 16]. Directly mitigating the bias may lead to over-mitigation and unsatisfying results.

**Long-tail Phenomenon**    Long-tail phenomenons happen frequently in the Baidu-ULTR as shown in Fig. 2. We selectively emphasize two important long-tail phenomenons as follows. (1) Most retrieved documents are irrelevant to the user query as shown in the expert annotation. Over 50% of documents are annotated as irrelevant with the user query while only 1.8% of documents are annotated as perfect. (2) The user query also shows the long-tail distribution. The top 10% high-frequency queries occupy over 60% of search logs. In Tab. 7, the performance of ULTR algorithms on the high-frequency queries is much higher than the tail queries.

**Mismatch between Training and Test**    Due to the storage limitation in the online system, we only record the displayed pages in the ranking system, which is usually the top-10 results. However, in the evaluation set, we record the top-30 documents and further cover the top thousands of results from the retrieval stage by selecting the document with an interval of 30. The sample results indexes are {30, 60, 90, 120, 150, ..., 990}. This challenge widely exists in the practical scenario, an arbitrary number of documents can be retrieved into our ranking procedure. The number of documents for training and testing can seldom be exactly aligned. The data mismatching is an out-of-distribution challenge for ranking algorithms.

### 5.2  Research Topics

In this subsection, we introduce three advanced research topics with great practical value. Baidu-ULTR first enables the academic studies of those topics in ULTR.

**Pre-training models for Ranking.**    The pre-training learning paradigm has been proven to be beneficial for improving downstream task performance. However, pre-training for ranking has not been well explored yet. In Baidu-ULTR, a large corpus of queries and documents has been provided,

which opens the opportunity for exploring the pre-training task for LTR. Moreover, finetuning on the downstream task also plays an important role in the learning paradigm. How to fine-tune the pre-training model with biased user feedback can also be a promising research direction.

**Causal Discovery.**   Due to missing real implicit feedback in the existing datasets, existing methods follow the paradigm of making assumptions about the clicking model and verifying the assumption in a simulation experiment, which might not be realistic. With Baidu-ULTR, we are capable of discovering the clicking model from the dataset and extracting the relevance model from the casual discovery model [12].

**Multi-task Learning.**   Different tasks can benefit each other by giving complementary information or counteracting task-independent information, such as CTR and CVR prediction [20] in the recommendation. With sufficient user behaviors, Baidu-ULTR is suitable for studying the benefit provided by the multi-complementary tasks, such as dwelling time prediction, which might promote this direction.

### 5.3   Data Limitations

In this subsection, we introduce three potential limitations of Baidu-ULTR that should be noticed when utilizing Baidu-ULTR.

**Inconvenient Model Sharing.**   Due to the privacy issue, the original text is translated into token ids with a private dictionary. Furthermore, to avoid losing the semantic features, we provide a set of high-frequency words with the combination of tokens in Baidu-ULTR. Though the desensitized original text is beneficial for extracting all kinds of semantic features for LTR easily, commonly used pre-trained language models trained with large corpus such as BERT and ERNIE cannot be used directly. We have to retrain models solely with the Baidu-ULTR corpus. For easy usage, we provide pre-trained language models as described in Section 4.

**User Diversity.**   Baidu-ULTR is collected from the query sessions in the Baidu search engine, which majorly targets on the Chinese community. This may induce the lack of user diversity and language diversity. Statistically speaking, there are about 97% queries in Chinese and about 2% queries in English.

**Biases in Expert Annotation**   Though we have designed a detailed 80-page guideline for annotation, designing the expert annotation guideline is a non-trivial task. There are still two potential biases in the expert annotation dataset: (1) To some extent, improving on the debiasing algorithms might not reflect on the test dataset since the 5-level annotation is typically more coarse than user feelings on the results. (2) In some special queries, the guideline might conflict with the relevance judgment from users' feedback since there is no guideline suitable for all queries. For example, in medical queries, the authority may be more important since the users are more concerned about correctness of the documents. However, in the general guidelines, the timeless document is judged as more relevant than the authoritative document. Though, we have an 80-page guideline that considers all the known exceptions, we cannot guarantee to cover all the special cases.

## 6   Conclusion

This paper introduces a large-scale web search dataset, Baidu-ULTR dataset. It includes large-scale web search sessions for training with real-world user feedback, and 7,008 expert annotation queries (397,572 query document pairs) for evaluation. The Baidu-ULTR contains adequate display information and rich types of user feedback. Moreover, the raw textual feature is also provided after desensitization, which enables the utilization of more advanced language models. A set of well-trained language models is also provided. We systematically introduce each component in the Baidu-ULTR and the instruction on how to conduct the dataset. Empirical studies show the limitations of the existing ULTR algorithms and the great potential to develop new algorithms with Baidu-ULTR. We also carefully consider the broader impact from various perspectives such as fairness, security, and harm to people. No apparent risk is related to our work.

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
