# OpenReview forum: "A Large Scale Search Dataset for Unbiased Learning to Rank"
_NeurIPS.cc/2022/Track/Datasets_and_Benchmarks — NeurIPS 2022 Datasets and Benchmarks _

### Official Review · Reviewer_tDsJ · 2022-07-24
**A good dataset for unbiased learning to rank research**

**Rating:** 7
**Confidence:** 4
**Correctness:** See "Summary And Contributions"
**Clarity:** The paper is well written.

**Strengths:**

See "Summary And Contributions"

**Weaknesses:**

See "Summary And Contributions"

**Additional Feedback:**

See "Summary And Contributions"

**Documentation:**

No problem with documentation.

**Ethics:**

No apparent ethical and social implication risk should be related to this work

**Relation To Prior Work:**

See "Summary And Contributions"

**Summary And Contributions:**

This paper introduces the Baidu-ULTR dataset for unbiased learning to rank.

The contributions of this work include:
1. It introduces original text in the form of token ids for queries and documents so that the pre-trained language models can be used
2. It provides more diverse display information in addition to the position information, which is an improvement over previous works
3. It also offers rich user behaviors like dwelling time, displayed time, etc., in addition to the click behavior
4. Compared with previous works, this is a large-scale dataset, with real user feedback and expert annotations. The long-tailed query research is also supported by design.
5. More tasks can be done in addition to the click modeling, e.g., dwelling time prediction, etc.

In addition to the contributions, the strengths of this work include:
1. Unbiased learning to rank is an important research topic, I believe this dataset can greatly accelerate this direction's development and bring benefits to the broader research community
2. The dataset is publicly available, baselines, and pre-trained models are available

The weaknesses of this work include:
1. As mentioned in the paper, due to the privacy issue, the texts are transformed into token ids, making the most commonly used pre-trained models not applicable to this dataset, thus limiting the impact of this dataset. Also, it makes the data analysis hard, as a result, the following researchers can only work on the algorithm or model side
2. Also as mentioned in the paper, there is a mismatch between training and test data distributions.

---

> ### Author Response · Authors · 2022-08-11
> **Response to reviewer tDsJ**
>
> We appreciate this reviewer’s overall positive assessment of our contributions and are grateful for your suggestion. We have already updated the new revision based on your suggestions.
>
> ---
>
> **W1: Due to the privacy issue, the texts are transformed into token ids, making the most commonly used pre-trained models not applicable to this dataset.**
>
> **R:** Inconvenient model sharing has the following two limitations:
>
> - We can not utilize the open source large-scale pretrained language models like BERT, and ERNIE.
> - The model pretrained on our dataset, cannot be utilized in other datasets. However, the pre-trained task can be shared between these models.
>
> The main reason for this limitation is the privacy protection constraint. We are not available to expose the original text feature in our dataset. The original texts are mapped into token uids with a private dictionary.
> BERT models cannot be used because of missing original texts. Also, as the dictionary is private which may be different in different datasets, a word may have different uids in different datasets. The pretrained model on our dataset can not be utilized to other datasets.
>
> Nonetheless, we are the first to provide the uids instead of pre-extracted features like BM2.5. Our Baidu-ULTR dataset first gives the opportunity of the large-scale language models into the academic research of ULTR.
>
> To solve this limitation, we provide sets of pretrained language model for easy usage.
> For the future direction, we aim to alleviate this limitation by conducting more related datasets with the same privacy word-to-ids dictionary.
>
> ---
>
> **W2: There is a data mismatch between training and test data distributions.**
>
> **R:**  After discussion, we find mismatching between training and test should be a limitation but an out-of-distribution challenge. New description can be found in the data challenge subsection of our new paper revision.
>
> We firstly give a concrete description of the mismatch between training and test. For queries in the training set, we record the displayed documents for each query, while the top-30 + 30 from 30-1000 results documents are recorded in the test set. A more detailed description of the data collection can be found in the response to W1.
>
> Data mismatching always exists in the real-world scenario, since an arbitrary number of documents can be retrieved into our ranking procedure.  It is more like an out-of-distribution (o.o.d) challenge rather than a limitation.
>
>
>
> ---
>
> Thank you again for your constructive reviews. Hope that our response can address your concerns. We will feel grateful if you could boost our paper.

---

### Official Review · Reviewer_8nET · 2022-07-26
**This is a great paper with lots of contributions. The proposed datasets could dramatically benefit other researchers' work.**

**Rating:** 9
**Confidence:** 4
**Clarity:** The paper is well written.

**Strengths:**

First, this paper has the contributions of providing the original semantic
feature and a pre-trained language model, sufficient display
information and rich user feedback on search result
pages. Besides, this dataset is general enough, and can be used in other research areas. Also, the paper provides enough metric results.

**Weaknesses:**

The amount of expert annotations queries is far less than the amount of search session. The authors did not provide enough explanation why 7,008 expert annotations are sufficient for 1.2 billions searching sessions. Besides the expert annotation distribution is not balanced. The perfect only takes 1.8% of total expert annotations. It will be better if the authors could provide more expert annotated queries with balanced distribution.

**Additional Feedback:**

Generally, this is a great paper with remarkable contribution and enough metric results.

**Correctness:**

The dataset is constructed in a sound way. The evaluation methods is appropriate.

**Documentation:**

There is sufficient detail on data collection and organization. For the benchmarks, there is also sufficient detail to support reproducibility.

**Ethics:**

No ethical concern at this moment.

**Relation To Prior Work:**

Yes, this paper clearly discussed previous the disadvantages of previous similar datasets, and clear shows what they provides.

**Summary And Contributions:**

As the paper states in its abstract, the proposed dataset provides original semantic features and pre-trained language model. It also provides sufficient display info and rich user feedback.

---

> ### Author Response · Authors · 2022-08-11
> **Response to reviewer 8nET**
>
> We appreciate this reviewer’s inspiring positive assessment of our contributions and are grateful for your suggestion. We have already updated the new revision based on your suggestions.
>
> ---
>
> **W1: lack of enough explanation why 7,008 expert annotations are sufficient for 1.2 billion searching sessions.**
>
> **R:**  Notice that, the 7,008 expert annotations are the 7,008 expert annotation queries.  We actually have 367,262 annotated query-document pairs which is already a large enough validation.  We are now conducting an additional 7,000 expert annotation queries, it will be publicly available as soon as possible.
> Then our expert annotation set will become the largest one in the existing ULTR dataset. We think that 7,008 expert annotation queries are sufficient for the ULTR task which can achieve  statistically significant results.
>
>
>
> ---
>
> **W2: The expert annotation distribution is not balanced. The perfect only takes 1.8% of total expert annotations.**
>
> **R:** Thanks for your good question, a detailed discussion has been provided in the expert annotation, Section experiment in our paper revision.
>
> The key points are as follows: We first add a description of the guideline for expert annotation.
>
> | Label         | Guideline                                                    |
> | ------------- | ------------------------------------------------------------ |
> | 0 (bad)       | Useless or outdated documents that do not meet the requirements at all. |
> | 1 (fair)      | Helpful to some extent but deficient in authority, timeliness document. |
> | 2 (good       | Meet the requirement of the query.                           |
> | 3 (excellent) | Meet the requirement of the query and timeliness document.   |
> | 4 (Perfect)   | Meet the requirement of the query, timeliness, and authoritative document. |
>
> The perfect label is only 1.8% in all expert annotations since a perfect will be only given to the destination page of a navigational query.
>
> It is natural that a bad label is much more than a perfect label. The reason is that the irrelevant documents are usually the majority for long-tail queries, e.g., Does the China Tobacco Bureau have any temporary job position?  And the long-tail queries are also the majority for the search engine.
>
>
>
> ---
>
> Thank you again for your constructive reviews. Hope that our response can address your concerns. We feel grateful for your appreciation.

---

### Official Review · Reviewer_def7 · 2022-07-26
**A new database for ULR**

**Rating:** 5
**Confidence:** 2
**Clarity:** The paper is well written and easy to…

**Strengths:**

The main interest of this paper is to provide a database for unbiased learning to rank.

**Weaknesses:**

The contribution on the collected data remains for me rather weak. The annotation by experts is quite basic. The use of this database, even if it is made available, will probably not be easy.

**Additional Feedback:**

It could be interesting to propose a less strict annotation of the experts, for example a fuzzy annotation.

**Correctness:**

For this dataset, queries are randomly sampled from search sessions of the Baidu search engine in April
2022. Expert annotation was made with 5 relevance labels of (bad, fair, good, excellent, perfect). Half are bad.

**Documentation:**

The appendix gives a good description of the database.

**Ethics:**

There is no mention on the personal data in the database.

**Relation To Prior Work:**

The link with other database is good.

**Summary And Contributions:**

This paper presents a new real database for ULR based on Baidu chinese application. This database is large in size. It is compared to other databases of this type which are already numerous.
The database is available free of charge. The textual part is in Chinese, which does not facilitate its use.

---

> ### Author Response · Authors · 2022-08-11
> **Response to reviewer def7**
>
> We appreciate this reviewer’s suggestion. We have already updated the new revision based on your suggestions.
>
> ---
>
> **W1: The contribution of the collected data remains is rather weak.**
>
> **R:** Overall speaking, our Baidu-ULTR dataset not only opens the door to multiple research topics in ULTR but also mitigates the gap between practice and academic study.
> We would like to highlight our key contributions of our Baidu-ULTR as follows.
>
> -  We first provide the raw text feature after desensitization instead of the pre-processed semantic feature like BM2.5.  Moreover, Baidu-ULTR is the largest dataset, with 1.2 billion training queries. This enables the study of the large-scale language model in the ULTR domain.
> -  We provide rich user behavior, while other datasets only provide click data. This enables the study of both multi-task learning and optimizing user engagement in the ULTR domain.
> -  We provide rich displayed features, while other datasets only provide the position data. This raises a large room for the study of biases introduced by different displayed feature
>
> ---
>
> **W2: The annotation by experts is quite basic.**
>
> **R:** Thanks for your great suggestion. The 5-level annotation has been widely adapted in learning to rank[1,2]. We have added details on the annotation guideline in our paper revision. The guideline rule is illustrated as follows.
>
> Each document is annotated by more than three experts. Only the label that most expert annotators agree on will be selected as the final label. It ensures the high quality of annotation.
>
> ---
>
> **W3: The use of this database, even if it is made available, will probably not be easy since Expert annotation was made with 5 relevance labels of (bad, fair, good, excellent, perfect). Half are bad.**
>
> **R:** Thanks for your great suggestion, we have added details on why half labels are bad, which is actually a normal phenomenon in a ranking system. The reason is that the irrelevant documents are usually the majority for long-tail queries, e.g. Does the China Tobacco Bureau have any temporary job position?  And the long-tail queries are also the majority for the search engine.
>
>
>
> ---
>
> Thank you again for your constructive reviews. Hope that our response can address your concerns. We will feel grateful if you could boost our paper.
>
>
> [1] Chapelle O, Chang Y. Yahoo! learning to rank challenge overview[C]//Proceedings of the learning to rank challenge. PMLR, 2011: 1-24.
>
> [2] Liu T Y. Learning to rank for information retrieval[J]. Foundations and Trends® in Information Retrieval, 2009, 3(3): 225-331.

---

### Official Review · Reviewer_3wwS · 2022-07-26
**nice contribution for building large scale search dataset with useful features, from certain type of users**

**Rating:** 6
**Confidence:** 3

**Strengths:**

- The search dataset covers a large range of web searches and queries
- The authors extracted a different kind of information from these web pages
- The authors tried to utilize the literature loss equations but adapted for the user’s implicit feedback for unbiased learning.
- The authors compared different benchmarks for showing the performance of the dataset using unbiased learning.
- I liked that they showed the tail queries performance and showed how the performance degrades for such queries.

**Weaknesses:**

- The work is based on extracting pages from Baidu search engine, which targets mainly the Chinese world, this causes the dataset to lack diversity and lack different users' queries.
- The work gives high importance to the users' clicks in the introduction and building of the dataset, however in the discussion and the results we can not reach this importance in the end, the naive benchmark with the standard features is performing really well. This needs more illustration how these features can impact the performance.
- Similarly, the discussion of the performance tables needs more illustration, on how the display information and user behavior impact the performance, also more illustration of the different accuracy results of DCG@1, DCG@3, ....etc.


**Additional Feedback:**

- The dataset is very useful to the Chinese research community, however, to extend its usefulness to further communities, it would be great if we have diverse search queries from different engines that do not focus on the Chinese user. For other research communities, the usefulness is questionable.

- The paper is a nice contribution and opens research questions about the usefulness of added features.




**Clarity:**

The paper is generally well written, except for the sections on results and discussion, which need paraphrasing and extra illustration in some parts.
Also, the arrangement of the experimental framework is a bit confusing, I would move the evaluation metrics to the end and speak first about the models and the baselines.
As far as I understand the annotated data can be used for evaluation but is all the data annotated with enough external evaluators?
The tables did not illustrate this well in my point of view.

**Correctness:**

The claims are correct, and the construction of the dataset seems sound.
There are external annotators to annotate data to be useful for evaluation.
The experimental design seems appropriate when I revised the experimental design in this area. However, I am not quite familiar with the experimental setup in this area, this should be revised by another reviewer.

**Documentation:**

The Github link of the models has some reproducibility details.
However, I could not examine the data because all the folders in the link https://github.com/ChuXiaokai/baidu_ultr_dataset/tree/main/data seem empty.
I read that for online storage there are some limitations, but if there is a limitation, how are you planning to open source the data?

**Ethics:**

There are no ethical concerns to my knowledge.

**Relation To Prior Work:**

Mainly added features and a much larger dataset, also an enhancement to the loss function used for training the model for better unbiased learning.
Here, too, I would like to refer to another reviewer for the writing of the adaptation of the loss function and if this adaptation was essential and useful.

**Summary And Contributions:**

The paper proposes a large scale dataset of search queries from Baidu search engine, they added behavioral features and page presentation features.
The contributions are as following:
- largest-scale search dataset
- added behavioral features
- added page presentation features
- expert annotations
- comparison of various ranking benchmarks

---

> ### Author Response · Authors · 2022-08-11
> **Response to reviewer 3wwS (1/2)**
>
> We appreciate this reviewer’s overall positive assessment of our contributions and are grateful for your suggestion. We have already updated the new revision based on your suggestions.
>
> ---
>
> **W1: The dataset may lack diversity and different user behavior since it mainly targets the Chinese world.**
>
> **R:** Thanks for your effort to point out this valuable limitation. We have included this valuable point in the limitation discussion. Notice that except for queries in Chinese, there also exists about 2% English queries which indicate the diversity in our dataset.
>
> I think this limitation is very difficult to solve in practice. The only solution is to collect search queries from different engines. However, this is impractical due to business competition and user privacy concerns.
>
> ---
>
> **W2: Inconsistency on the introduction and experiments, experiments do not verify the importance of debias on click.**
>
> **R:** This is a very good question. The following discussion has been included in the experiment section of our new revision.
>
> In the introduction part, we introduce the click can be biased and mitigate the bias on the click could receive unbiased better performance. This is a general perspective on the existing ULTR algorithms.
> However, in the experiments part, the naive algorithm directly utilizing the click for training
> shows comparable performance to the baseline.
>
> The major reason is that baseline methods only take the position-related bias into consideration without utilizing other display features. Moreover, they only include the click data as the supervised signal without any other user behavior data.  Therefore, it is no wonder that those methods can not perform well since baseline methods only consider one particular bias.  Accordingly, there is a large room for improvement for considering biases on multiple display features and user behavior together.
>
> Another potential reason for the unsatisfying performance can be the gap between synthetic data and real-world data. Currently, most academic datasets do not provide real-world click information, which utilizes synthetic data instead. Baidu-ULTR provides the real-world click from a well-performed search engine. It can be very different between the synthetic data and our real-world scenario. The click may not have the same simple and strong bias as the assumption in generating synthetic dataset. Directly mitigating the bias may lead to over-mitigation and unsatisfying results in practice.
>
> The above discussion also points out the practical value of our Baidu-ULTR dataset, which mitigates the gap between academic and industry scenarios.
>
> ---
>
>
> **W3: Similarly, the discussion of the performance tables needs more illustration, on how the display information and user behavior impact the performance, also more illustration of the different accuracy results of DCG@1, DCG@3, ....etc.**
>
> **R:** The response is similar to the above one. The existing baseline methods do not take display features other than position into consideration.  Those discussions should be left for further algorithm designs on our Baidu-ULTR dataset
>
> ---
>
> **W4: the consideration on the annotation quality.**
>
> **R:** Thanks for your good question, We provide more detailed guideline on the annotation procedure in the dataset section and further discuss the potential limitations in the discussion section.
> The key points are as follows.
>
> The description of the guidelines for expert annotation is as follows. Each document is annotated by more than three experts. Only the label that most expert annotators agree on will be selected as the final label. It ensures the high quality of annotation.
>
> | Label         | Guideline                                                    |
> | ------------- | ------------------------------------------------------------ |
> | 0 (bad)       | Useless or outdated documents that do not meet the requirements at all. |
> | 1 (fair)      | Helpful to some extent but deficient in authority, timeliness document. |
> | 2 (good       | Meet the requirement of the query.                           |
> | 3 (excellent) | Meet the requirement of the query and timeliness document.   |
> | 4 (Perfect)   | Meet the requirement of the query, timeliness, and authoritative document. |
>
> Bias may also come from discrete labeling since we only have five labels. However, user satisfaction is a more complex behavior which can only be coarsely described by those five labels.
>
> Notice that in our guideline, the timeless document is judged as more relevant than the authoritative document.  However, this may induce bias in some special cases, for example, queries about medical treatment, the authority may be more important since the users are more concerned about correctness of the documents.

---

> > ### Author Response · Authors · 2022-08-11
> > **Response to reviewer 3wwS (2/2)**
> >
> >
> > **Document problem: The reviewer cannot examine the data.**
> >
> > **R:** Since it is a large dataset which over the Github limitation. We open source our data on Google Drive.
> > The links are available at https://drive.google.com/drive/folders/1Q3bzSgiGh1D5iunRky6mb89LpxfAO73J?usp=sharing, and https://drive.google.com/file/d/1hdWRRSMrCnQxilYfjTx8RhW3XTgiSd9Q/view?usp=sharing
> > , for training and test set, respectively.  We also provide those URLs in the readme of our GitHub repository and our new dataset homepage.
> >
> >
> >
> >
> >
> > ---
> >
> > Thank you again for your constructive reviews. Hope that our response can address your concerns. We will feel grateful if you could boost our paper.

---

### Official Review · Reviewer_u8QG · 2022-07-28
**Baidu-ULTR: A good work to promote the development of the field of ULTR**

**Rating:** 7
**Confidence:** 4

**Strengths:**

1.The idea of proposing a large-scale search dataset Baidu-ULTR from industry for unbiased learning to rank is particularly meaningful and necessary.
2.This paper presents the design principle of Baidu-ULTR and the performance of benchmark ULTR algorithms on the new dataset in detail, which has the potential to become the gold benchmark in the field of ULTR.
3.In general, this paper is well-organized and is a good work that can greatly promote the development of the field of ULTR.

**Weaknesses:**

1.The presentation of the paper should be improved. The expressions in current version are not natural and elegant enough, and it would be better to improve it with the help of an English native speaker.
2.The layout of the paper should be optimized, especially the tables and figures in current version (e.g., compactness, color, etc.).
3.Because Baidu-ULTR is a large scale real-world dataset from industry, the balance of open source and privacy protection requires extra care. And some details of dataset collection and data pre-processing, including data desensitization, should be more clear.
4.As paper claims, data limitations, including inconvenient model sharing and mismatch between training and test, should be discussed more.

**Additional Feedback:**

In general, this paper is a good work that can greatly promote the development of the field of ULTR, and the proposed dataset Baidu-ULTR has great potential to become the gold benchmark in this field.

**Clarity:**

In general, this paper is well-organized and well-written. And it would be better to improve the presentation of it with the help of an English native speaker.

**Correctness:**

From my perspective, the dataset Baidu-ULTR is constructed in a sound way and the performance of benchmark ULTR algorithms on it is correct.

**Documentation:**

The documentation has been given in detail. The proposed dataset Baidu-ULTR and the corresponding benchmark have been described in as much detail as possible.

**Ethics:**

The ethical concerns of the proposed dataset Baidu-ULTR have been discussed thoroughly. And the balance of open source and privacy protection requires extra care.

**Relation To Prior Work:**

As far as I can tell, this paper has clearly discussed how its proposed Baidu-ULTR differs from previous contributions.

**Summary And Contributions:**

This paper proposes a large-scale unbiased learning to rank dataset for mobile web search, Baidu-ULTR, which randomly sampled 1.2 billion searching sessions from the largest Chinese search engine Baidu, and 7,008 expert annotated queries for validation and test. Baidu-ULTR provides: (1) the original semantic feature and a pre-trained language model for easy usage; (2) sufficient display information for enabling the comprehensive study of different biases; and (3) rich user feedback on search result pages (SERPs) for user engagement optimization and promoting the exploration of multi-task learning in ULTR. In addition, this paper presents the design principle of Baidu-ULTR and the performance of benchmark ULTR algorithms on the new data resource, favoring the exploration of ranking for long-tail queries and pre-training tasks for ranking.

---

> ### Author Response · Authors · 2022-08-11
> **Respone to reviewer u8QG**
>
> We appreciate this reviewer’s overall positive assessment of our contributions and are grateful for your suggestion. We have already updated the new revision based on your suggestions.
>
>
>
> ---
>
> **W1: The presentation of the paper should be improved.  W2: The layout of the paper should be optimized**
>
> **R:**  We have checked and polished our paper. We hope that the new revision can provide a more satisfying reading experience. If there still remains any consideration, please kindly let us know.We are very happy to make a further revision in light of your great suggestions.
>
> ---
>
>
> **W3: more details should be provided on dataset collection and data pre-processing, including data desensitization.**
>
> **R:** Thanks for your great suggestion.  We have added more details on the dataset collection, and data pre-processing in our revision. Here, we summarize some key points in our new revision for a better understanding.
>
> - We add the collection details on the documents of the Large Scale Web Search Session. It collects all the documents displayed to the user.
>   It usually contains just 10 documents since users usually only stay on the first page.
> - We add the collection details on the documents of the expert annotation dataset. We record the top-30 documents from the retrieval model and sample documents over the top thousands of results from the retrieval stage (the documents ranked at position \{30, 60, 90, 120, 150, ..., 990\}). They are provided to measure how the model can distinguish the document, which is disruptive to the user experience.
> - A detailed expert annotation guideline is also provided in our paper.
>
> For the data preprocessing, there is no specific design except for the data desensitization. Data desensitization on the textual feature aims to protect the user's privacy. It transforms the original raw textual feature into the token unique identifier. For example, the word "we" will change to a unique identifier "5". This transformation is conducted based on a privacy dictionary. We further emphasize this point in our paper revision.
>
>
>
>
> ---
>
> **In weakness 4**, the reviewer points out the data limitation on (1) inconvenient model sharing (2) mismatch between training and test.
> Notice that, we find that the mismatch between training and test should not be a limitation, but a practical out-of-distribution challenge.
> We will respond to those two concerns respectively as follows.
>
> **W4.1: More discussion should be provided on inconvenient model sharing.**
>
> **R:** Inconvenient model sharing has the following two limitations:
> (1) We cannot utilize the open source large-scale pretrained language models like BERT, and ERNIE.
> (2) The model pretrained on our dataset cannot be utilized in other datasets. However, the pre-trained task can be shared between these models.
>
> The main reason for this limitation is the privacy protection constraint.  We are not available to expose the original text feature in our dataset.  The original texts are mapped into token uids with a private dictionary.
> BERT models cannot be used because of missing original texts. Also, as the dictionary is private which may be different in different datasets, a word may have different uids in different datasets.
> The pretrained model on our dataset can not be utilized to other datasets.
>
> Nonetheless, we are the first to provide the uids instead of pre-extracted features like BM2.5.
> Our Baidu-ULTR dataset first gives the opportunity of the large-scale language model into the academic research of ULTR.
>
> **W4.2: More discussion should be provided on mismatching between training and test.**
>
> **R:** After discussion, we find mismatching between training and test should not be a limitation but an out-of-distribution challenge. The new description can be found on the data challenge subsection of our new paper revision.
>
> We firstly give a concrete description of the mismatch between training and test. For queries in the training set, we record the displayed documents for each query, while the top-30 + 30 from 30-1000 results documents are recorded in the test set. A more detailed description of collection can be found in the response to W1.
>
> Data mismatching always exists in the real-world scenario, since an arbitrary number of documents can be retrieved into our ranking procedure.  Data mismatching is more like an out-of-distribution (o.o.d) challenge rather than a limitation.
>
>
>
> ---
>
> Thank you again for your constructive reviews. Hope that our response and additional experiment results can address your concerns. We will feel grateful if you could boost our paper.

---

### Official Review · Reviewer_UJkQ · 2022-07-28
**Good paper, large dataset.**

**Rating:** 8
**Confidence:** 4
**Clarity:** The paper is well structured and easy…

**Strengths:**

(1) The dataset itself is quite large for a search dataset. 1.2B sessions and 7k+ expert annotations. This would be very useful for researchers not working at giant search engine companies.
(2) Most search papers/datasets focus on solely the rank part of search result pages. With modern search evolving to detailed abstracts and rich multimedia content within the results page, it is important to understand their effect on user engagement.


**Weaknesses:**

The paper does not have many weaknesses. It is a large scale search dataset that would be pretty useful. The only concern I have is with the expert annotations and what kind of biases raters may have introduced while providing the relevance labels. I think a brief section about the rater guidelines and potential problems they could introduce into the dataset would be useful. With the data being anonymized we may never really know if experts tend to favor certain kinds of documents over others.

**Additional Feedback:**

Sentences do have a lot of grammatical mistakes but they don’t affect the overall flow of information. Would urge the authors to reword some of them if possible, for ex:
(1) “BERT cannot be exploited due to the missing of the original text” -> “BERT models cannot be used because of missing original text”
(2) “missing the displayed abstract of documents for analyzing the click necessary bias” -> “missing abstract information useful for analyzing click bias”
(3) “vogue of deep learning” ? does this mean “rise of deep learning”
(4) “consider the border impact” -> broader*


**Correctness:**

The data collection process and baseline methods seem to be correct. In particular Fig. 3, 4 are typical of search datasets and the authors provide DCG scores with a few different ULTR methods.

**Documentation:**

Dataset and baselines are well described. Paper includes a github link which contains data and scripts to get started with training baseline models.

**Ethics:**

No ethics questions. All of the released data is fully anonymous (token ids from a private dictionary). So the dataset is still very useful but reveals nothing about the user or their click preferences.

**Relation To Prior Work:**

Prior work is well discussed. I agree with the authors, there are no/few datasets out there that contain detailed SERP level info along with user engagement metrics. Also most works before do not release query/document text with user feedback or are too small.

**Summary And Contributions:**

The paper provides a large scale dataset for search containing samples from Baidu’s search logs. The dataset contains a lot of new information about aspects of the search results page or user interaction that were previously ignored by other papers or were too small to train large transformer models. The information is also released in an anonymized way reducing the risk of user privacy exposure.

---

> ### Author Response · Authors · 2022-08-11
> **Respone to reviewer UJkQ**
>
> We appreciate this reviewer’s overall positive assessment of our contributions and are grateful for your suggestion. We have already updated the new revision based on your suggestions.
>
> ---
>
> **W1: The rater may introduce bias into the expert annotation.**
>
> **R:** Thanks for your constructive suggestions.
> We first add a description on the guideline for expert annotation as follow.
>
> | Label         | Guideline                                                    |
> | ------------- | ------------------------------------------------------------ |
> | 0 (bad)       | Useless or outdated documents that do not meet the requirements at all. |
> | 1 (fair)      | Helpful to some extent but deficient in authority, timeliness document. |
> | 2 (good       | Meet the requirement of the query.                           |
> | 3 (excellent) | Meet the requirement of the query and timeliness document.   |
> | 4 (Perfect)   | Meet the requirement of the query, timeliness, and authoritative document. |
>
> Bias may also come from the discrete labeling since we only have five labels.
> However, user satisfaction is a more complex behavior which can not only be coarsely described by those five labels.
>
> In some special queries, the guideline might conflict with the relevance judgment from users' feedback since there is no guideline suitable for all queries. For example, in medical queries, the authority may be more important since the users are more concerned about correctness of the documents. However, in the general guideline, the timeless document is judged as more relevant than the authoritative document. Though, we have an 80-page guideline that considers all the known exceptions, we can not guarantee to cover all the special cases.
>
> A more detailed description is provided in the data limitation section in the new revision.
>
>
>
> ---
>
> **W2: the grammatical mistakes in our paper.**
>
> **R:** We have checked and polished our paper. We hope that the new revision can provide a more satisfying reading experience.
> If there still remains any consideration, -please kindly let us know.
> We are very happy to make a further revision in light of your great suggestions.
>
>
>
> ---
>
> Thank you again for your constructive reviews. Hope that our response can address your concerns. We feel grateful for your appreciation.

---

### Author Response · Authors · 2022-08-11
**General response**

We thank all reviewers for their insightful comments and suggestions. We are particularly encouraged by the reviewers’ feedback. We have made a heavy revision to our paper according to the reviewer's constructive suggestions. Additionally, we construct a new homepage website for our Baidu-ULTR dataset, which can be found at https://huanhuqueyue.github.io/baidu_ultr_page/index.html. Below we summarize some key modifications in this revision:

- A more detailed description of the expert annotation.
  - The description of expert annotation guidelines is provided in Tab. 3.
  - The reason why the annotation label is imbalanced. Briefly speaking, the reason is that the irrelevant documents are usually the majority for long-tail queries, e.g., "Does the China Tobacco Bureau have any temporary job position?" Irrelevant documents are the majority of the retrieval documents. Unfortunately, the long-tail queries are also the majority for the search engine (referred to Fig. 2(a)).

- More details on the collection of documents
  - We provide more details on the document collection.
  - More discussions and emphasis on data desensitization for privacy protection.
- More explanation and analysis of the experimental results.
  - We emphasize the limitation on the baseline methods.
  - We provide more explanations on why those baseline methods cannot work well in the Baidu-ULTR dataset.

- A discussion subsection on the additional challenge in our dataset, including
  - biases in the real-world user feedback
  - Long-tail Phenomenon
  - Mismatch between the train and test dataset. Notice that, we consider this phenomenon as not a limitation, but a challenge that widely exists in the real-world ranking models.
- More discussions on the limitation including
  - User diversity: the data collected from the Baidu-ULTR dataset is mainly from the Chinese community.
  - Biases may be introduced in the expert annotation.


Moreover, We have carefully checked our paper typos and reorganized the presentation. If there still remains any consideration, Please kindly let us know. We are very happy to make a further revision in light of your great suggestions.  We will address comments by each of the reviewers individually.

---

### Meta-Review · Area_Chair_T1Dz · 2022-09-09

**Recommendation:** Accept
**Confidence:** 4

**Metareview:**

The reviewers seem to be in agreement on many aspects, including the strengths, the weaknesses, and the final assessment.

As strengths, the reviewers highlight the following points:
- the size of the dataset,
- the usefulness to the community

As weaknessess, the reviewers highlight the following points:
- the paper in layout and grammer
- potential bias towards languages and industry
- biases with respect to the annotaters

The reviewers provide scores ranging from 5 (although the reviewer concedes that they are not certain about their assessment) until 9 (rather confident), averaging at a 7. I agree with the aforementioned strong points, and do not see any showstoppers in the recurring weaknesses, so will therefore recommend acceptance of the paper.

---

### Decision · Program_Chairs · 2022-09-16

Accept